# Risk Factors of Venous Thromboembolism in Noncritically Ill Patients Hospitalized for Acute COVID-19 Pneumonia Receiving Prophylactic-Dose Anticoagulation

**DOI:** 10.3390/v14040737

**Published:** 2022-03-31

**Authors:** Francesco Poletto, Luca Spiezia, Chiara Simion, Elena Campello, Fabio Dalla Valle, Daniela Tormene, Giuseppe Camporese, Paolo Simioni

**Affiliations:** General Internal Medicine Unit, Department of Medicine, Padova University School of Medicine, 35138 Padova, Italy; francesco.poletto1@gmail.com (F.P.); luca.spiezia@unipd.it (L.S.); solidea.cs@gmail.com (C.S.); elena.campello@unipd.it (E.C.); fabio.dallavalle.1@unipd.it (F.D.V.); daniela.tormene@unipd.it (D.T.); giuseppe.camporese@aopd.veneto.it (G.C.)

**Keywords:** venous thromboembolism, COVID-19, prophylactic-dose anticoagulation

## Abstract

**Background**: Therapeutic/intermediate-dose heparin reduces the risk of thromboembolic events but increases the risk of major bleeding in patients hospitalized for acute COVID-19 pneumonia. **Objectives**: To prospectively assess the incidence of objectively proven venous thromboembolism (VTE) and identify predisposing risk factors in a cohort of hospitalized patients with acute COVID-19 pneumonia undergoing prophylactic-dose heparin. **Patients and methods**: All consecutive patients admitted for acute COVID-19 pneumonia to the General Internal Medicine Unit of Padova University Hospital, Italy between November 2020 and April 2021, and undergoing prophylactic-dose heparin, were enrolled. Demographic and clinical characteristics and laboratory and radiological findings were recorded on admission. Cases were patients who developed VTE during their hospital stay. Univariable and multivariable logistic regression analyses were used to ascertain the risk factors associated with developing in-hospital VTE. **Results**: 208 patients (median age: 77 years; M/F 98/110) were included; 37 (18%) developed in-hospital VTE during a median follow-up of 10 days (IQR, 4–18). VTE patients were significantly younger (*p* = 0.004), more obese (*p* = 0.002), and had a lower Padua prediction score (*p* < 0.03) and reduced PaO2/FIO2 ratio (*p* < 0.03) vs. controls. Radiological findings of bilateral pulmonary infiltrates were significantly more frequent in VTE patients than controls (*p* = 0.003). Multivariable regression showed that obesity (1.75, 95% CI 1.02–3.36; *p* = 0.04) and bilateral pulmonary infiltrates on X-rays (2.39, 95% CI 1.22–5.69; *p* = 0.04) were correlated with increased risk of in-hospital VTE. Conclusions: Obesity and bilateral pulmonary infiltrates on imaging may help clinicians to identify patients admitted to medical wards for acute COVID-19 pneumonia at risk of developing VTE despite prophylactic-dose heparin. Further studies are needed to evaluate whether the administration of therapeutic/intermediate-dose heparin may help prevent VTE episodes without further increasing the bleeding risk.

## 1. Introduction

Several studies on the severe acute respiratory syndrome coronavirus 2 (SARS-CoV-2) infection have reported different coagulation disorders, ranging from consumptive coagulopathy to severe hypercoagulability, in patients admitted to Intensive Care Units and Medical Wards [1,2,3]. This phenomenon of a hyperactive coagulation system is due to several pathophysiological mechanisms such as endothelial damage, severe inflammatory response due to the cytokine storm, and increased levels of procoagulant factors [4,5]. From a clinical standpoint, this activation of the coagulation cascade is associated with a procoagulant state that dramatically increases the risk of developing both arterial and venous thrombotic events in acute COVID-19 patients [6,7,8,9]. In particular, venous thromboembolism (VTE) has been reported as one of the most frequent complications, with an estimated incidence of up to 30% in patients hospitalized in medical wards [10,11,12,13]. Thus, hospitalized patients with acute COVID-19 pneumonia require heparin that has anticoagulant, anti-inflammatory, and antiviral activity [14,15,16]. At present, the optimal anticoagulation regimen to prevent VTE in patients admitted for acute COVID-19 is still a matter of debate. In a recent meta-analysis, Jorda et al. [17] showed that therapeutic/intermediate-dose vs. prophylactic-dose heparin reduces the risk of thromboembolic events but increases the risk of major bleeding. It is, therefore, of paramount importance to identify patients who may develop a thrombotic event despite receiving a prophylactic dose of heparin and so would benefit most from a therapeutic/intermediate-dose regimen. This would help reduce the overall bleeding risk in hospitalized patients with acute COVID-19. The aim of our prospective study was to evaluate the incidence of VTE in a group of patients consecutively admitted to the medical ward of Padua University Hospital for acute COVID-19 pneumonia and treated with prophylactic-dose heparin. We also aimed to identify potential predisposing risk factors linked to the development of thromboembolic events in these patients.

## 2. Materials and Methods

### 2.1. Study Design and Participants

We considered all patients admitted to the General Internal Medicine Unit of Padua University Hospital, Italy between November 2020 and April 2021, with a confirmed laboratory diagnosis of acute COVID-19 pneumonia according to the WHO interim guidance. All patients who received prophylactic-dose anticoagulation (i.e., subcutaneous enoxaparin 40 mg or fondaparinux 2.5 mg, once daily) upon admission were enrolled in the study. Exclusion criteria were refusal to participate in the study, therapeutic-dose anticoagulation (due to atrial fibrillation or acute venous thromboembolism), contraindications for anticoagulation (due to recent bleeding, renal insufficiency, or severe thrombocytopenia), and discontinuation of prophylactic-dose heparin during hospital stay. All patients were followed prospectively from admission until discharge or transfer to other departments. Cases were patients who developed objectively proven VTE—proximal and/or distal deep vein thrombosis (DVT) of the upper and/or lower limbs—and pulmonary embolism (PE). VTE was assessed by ultrasonography or computed tomography pulmonary angiography. Diagnostic tests were applied if thrombotic complications were clinically suspected. Controls were patients who did not develop venous thromboembolism during hospitalization.

The study was approved by the local institutional ethics committee (Ref: 3419/2015-652/AO/19) and conducted in compliance with the principles of the Declaration of Helsinki.

### 2.2. Data Collection

Epidemiological, demographic, clinical, laboratory, treatment, and outcome data were recorded by FP, LS, CS, and EC in an Excel data collection form. Cases and controls were compared in terms of age, sex, body mass index (BMI), comorbidities, blood group, Padua Prediction Score (PPS), Sequential Organ Failure Assessment (SOFA) score, Sepsis-Induced Coagulopathy (SIC) score, and PaO_2_/FIO_2_ ratio. In particular, we considered comorbidities such as arterial hypertension, diabetes, coronary artery disease, cerebrovascular disease, previous thrombotic events, active cancer, and chronic kidney disease. PPS is a clinical score that allows us to stratify thromboembolic risk in hospitalized medical patients and determine whether to initiate thromboprophylaxis [18]. The SOFA score is used to describe the extent of organ failure [19]. The SIC score predicts the likelihood of sepsis-induced coagulopathy using three variables: International Normalized Ratio, platelet count, and SOFA score [20]. The PaO_2_/FIO_2_ ratio measures the severity of the acute respiratory syndrome.

### 2.3. Laboratory Procedures

At admission, each enrolled patient provided throat swab specimens for real-time PCR COVID tests and the following blood analyses were performed: complete blood count, coagulation profile (including coagulation times, D-dimer, Fibrinogen, Antithrombin, Factor VIII, and Von Willebrand factor activity), serum creatinine, ferritin, interleukin-6 (IL-6), C-reactive protein (CRP), and procalcitonin. Whole blood (WB) clotting capability was measured by rotational thromboelastometry (ROTEM^®^-Instrumentation Laboratory, Milan, Italy) according to the manufacturer’s protocol and as previously reported [21]. Briefly, WB was put in a heated cup with a pin suspended and connected to an optical detector system. The firmer the clot, the higher the force opposed to the movement of the rotating pin. All viscoelastic tests were performed within 2 h of the blood collection and lasted 60 min. The extrinsic and intrinsic coagulation pathways were studied using EXTEM and INTEM assays, respectively. The role of fibrinogen in clot firmness was evaluated using the FIBTEM assay. The following ROTEM parameters were analyzed: (i) Clotting time (CT, s), corresponding to the time from the beginning of the coagulation analysis until an increase in amplitude of 2 mm. CT measures the initiation phase of the clotting process; (ii) Clot formation time (CFT, s), the time between an increase in amplitude of the thromboelastogram from 2 to 20 mm. CFT reflects the propagation phase of WB clot formation; (iii) Maximum clot firmness (MCF, mm), the maximum amplitude reached in the thromboelastogram. MCF correlates with the platelet count and function, as well as the fibrinogen concentration. Hypercoagulable ROTEM^®^ profile was defined as CT or CFT shorter and/or MCF higher than reference ranges. WB platelet function was measured by impedance aggregometry (Multiplate-Roche Diagnostics GmbH, Mannheim, Germany) according to the manufacturer’s protocol and as previously reported [21]. Briefly, activated platelets adhere and aggregate to two copper wires, resulting in an increased electrical resistance (i.e., impedance) between the two electrodes that is proportional to the number of adhered platelets. Results are reported as AUC (area under the curve) units. Agonists tested were: (i) adenosine-5′-diphosphate (ADP test), (ii) arachidonic acid (ASPI test) and (iii) thrombin receptor activating peptide (TRAP test). The multiplate platelet hyperaggregability profile was defined as higher AUC for an ADP, ASPI, or TRAP test than the upper limit of the reference range. Chest X-rays or CT scans were also performed on all enrolled patients.

### 2.4. Statistical Analysis

Continuous and categorical variables were presented as the median/interquartile range (IQR) and absolute numbers (percentage), respectively. Mann–Whitney U tests and χ^2^ tests were used to compare differences between cases and controls. Univariable and multivariable regression models were used to estimate relative risk (RR) with 95% confidence interval (CI) to evaluate the association between the incidence of in-hospital VTE despite prophylactic-dose heparin and demographic and clinical characteristics, as well as laboratory and radiological findings. The multivariable regression model was limited to variables with a *p*-value < 0.1. All statistical analyses were performed using PASW Statistics 17.0.2 (SPSS Inc., Chicago, IL, USA) for Windows.

## 3. Results

Demographic and clinical characteristics are given in Table 1.

Among the 261 patients considered for the study, 53 were excluded due to therapeutic-dose anticoagulation (*n* = 27), contraindications for anticoagulation (*n* = 14); lack of informed consent (*n* = 7), or discontinuation of prophylactic-dose heparin during hospitalization (*n* = 5). Two-hundred and eight patients were included in the study. During a median follow-up of 10 days (IQR, 4–18), 37 patients (18%) developed a thromboembolic event despite thromboprophylaxis. Fourteen patients (38%) developed catheter-related thrombosis. The type of VTE was PE with or without DVT in 3 (8%) patients, proximal DVT in 7 (19%), distal DVT in 12 (32%), and upper extremity DVT in 15 (41%). The median age was 77 (66–86) years; 110 patients (53%) were female and 98 (47%) were male. The median Body Mass Index (BMI) was 26 (23–31). The most frequent comorbidity was arterial hypertension (*n* = 131, 63%), followed by chronic kidney disease (*n* = 43, 21%) and diabetes (*n* = 39, 19%). The most prevalent blood groups in our study population were O (*n* = 85, 41%) and A (*n* = 83, 40%). The cases were significantly younger (71 (61–81) vs. 78 (69–86) years, *p* = 0.004), and more obese (BMI 29 (27–31) vs. 26 (23–29), *p* = 0.002) than the controls. The Padua Prediction Score was significantly lower in patients who developed VTE (3 (2–3) vs. 4 (3–6), *p* = 0.03). The PaO_2_/FIO_2_ ratio was lower in cases (258 (195–272) mmHg) than in controls (277 (232–320) mmHg, *p* = 0.03).

The laboratory and radiographic findings are reported in Table 2.

The hemoglobin concentration was slightly higher, albeit not statistically significantly so, in patients who developed VTE (128 (117–138) vs. 123 (107–137) g/L, *p* = 0.12). More controls presented hemoglobin concentration <120 g/L than cases (80 [47%] vs. 11 [30%], *p* = 0.06). No significant differences were observed in white blood cell count, platelet count, CRP, procalcitonin, and creatinine concentration. Serum ferritin was higher in cases, though not significantly (663 (476–1050) vs. 575 (335–894), *p* = 0.22). There were no significant differences in coagulation parameters: D-dimer levels were higher in cases (317 (197–1093) vs. 289 (161–603) ng/mL, *p* = 0.32), whereas coagulation times, fibrinogen, antithrombin, factor VIII, and von Willebrand factor activity were similar in both groups. No significant difference was observed in thromboelastometry and aggregometry parameters in cases and controls (Appendix A). We observed a significantly higher prevalence of bilateral pulmonary infiltrates on imaging in patients who developed thromboembolic events (68%) than controls (40%, *p* = 0.02).

The univariable and multivariable regression results are given in Table 3.

Elevated body mass index (>30) was confirmed as a risk factor for thromboembolic events (1.75, 95% CI 1.02–3.36; *p* = 0.04). Furthermore, both univariable and multivariable analyses revealed the association between thrombosis and bilateral pulmonary infiltrates on imaging (2.39, 95% CI 1.22–5.69; *p* = 0.04). A high Padua Prediction Score (>4) did not appear to increase the risk of developing in-hospital VTE in our study population (0.54, 95% CI 0.17–0.81, *p* = 0.04). No other clear associations or risk factors for the development of VTE were identified by univariable and multivariable regression analyses.

## 4. Discussion

Optimal thromboprophylaxis dosing in noncritically ill patients hospitalized for acute COVID-19 pneumonia in medical wards is still a matter of debate. The risk of thrombosis in these patients remains high (up to 30%) despite prophylactic-dose heparin. Therefore, our prospective study aimed to ascertain the possible risk factors for developing VTE despite prophylactic-dose heparin. We observed a high risk of VTE in our cohort, in line with the previous literature [7,10,11,12,13]. Several procoagulant mechanisms (i.e., platelet hyperactivation, increased coagulation factors, and endothelial cell damage) compounded by traditional VTE risk factors (i.e., central venous catheter, prolonged immobilization, and/or high-dose steroid treatment) may explain the high thrombotic risk despite prophylactic-dose heparin in these patients. However, the specific mechanism responsible for the inefficacy of low-dose heparin has not been described yet. Therefore, several trials were conducted to test higher (therapeutic and/or intermediate) heparin doses vs. prophylactic-dose heparin to reduce the risk of VTE [22,23,24,25]. Overall, these studies found that patients treated with higher doses of heparin had a lower risk of developing thromboembolic events but an increased risk of major bleeding [17]. Thus, it is paramount to identify any possible predisposing risk factors for the development of VTE despite prophylactic-dose heparin in order to limit the use of high-dose heparin to patients with the highest risk. We found that obesity and bilateral pulmonary infiltrates were independent risk factors for developing VTE, in agreement with previous studies in patients with acute COVID-19 pneumonia [26,27,28,29]. However, it bears noting that a recent meta-analysis by Lobbes et al. found that obesity was not associated with VTE in COVID-19 [30]. The different findings stem mainly from the heterogeneity of the study populations. The increased risk of developing VTE despite prophylactic-dose heparin observed in obese patients in our study supports current ISTH guidelines to administer higher doses of heparin in obese noncritically ill patients hospitalized for COVID-19 [31]. Another independent risk factor for developing VTE in our cohort was the presence of bilateral pulmonary infiltrates on imaging. We postulate that this may stem from microthrombi in pulmonary blood vessels as a result of direct endothelial damage caused by SARS-CoV-2 and by the cytokine storm leading to severe inflammatory response syndrome (SIRS) [32,33]. It bears noting that bilateral pulmonary infiltrates are a sign of more severe inflammation response due to pro-inflammatory cytokines that lead to a procoagulant state and, consequently, a higher thrombotic diathesis.

An important observation that emerged from our findings is that traditional risk factors for VTE such as advanced age, a personal history of thrombotic disease, or active cancer are not independent risk factors for the development of VTE in acute COVID-19 patients. Furthermore, the Padua Prediction Score (PPS), a clinical score that allows clinicians to identify medical patients at increased risk of thrombosis who may benefit from early thromboprophylaxis—the higher the score, the higher the risk—was significantly and unexpectedly lower in patients who developed VTE. These findings highlight the complexity and novelty of acute COVID-19 as a clinical entity with its own peculiarities, and whose pathophysiology remains largely unknown [34]. In particular, the mechanisms underlying the high VTE risk and thrombotic complications despite prophylactic-dose heparin are not fully elucidated yet. This highlights the urgent need for validated ad hoc VTE risk scores for patients hospitalized for acute COVID-19 pneumonia to tailor the optimal dose of heparin to the specific risk profile of each patient.

We observed a male/female ratio close to 1, whereas it is usually close to 3 in the literature. Although we have no clear explanation for this discrepancy, we can speculate that the particularly high median age (77 years) of the enrolled patients may have contributed to the increased prevalence of women.

We did not identify a laboratory parameter able to independently predict the risk of developing VTE. In our cohort, both traditional and whole blood coagulation parameters showed a markedly hypercoagulable profile without significant differences between cases and controls, in line with previous findings [2,3,35]. We also found similar D-dimer levels between cases and controls, unlike previous studies [36,37]. The heterogenicity and complexity of patients enrolled across studies make it difficult to compare the data.

One of the main limitations of our study is that it is a single-center study. While this allowed us to conduct a study with a small sample size, we presumably enrolled a more homogeneous group of patients, which may have reduced possible biases (e.g., therapeutic protocols for COVID-19 pneumonia; processing and analysis of samples). Moreover, for most of the parameters considered in our study, it was not possible to carry out repeated evaluations during hospitalization. Monitoring coagulation as well as the inflammation profile during the hospitalization could have provided more comprehensive data on predictive risk factors for VTE.

## 5. Conclusions

Obesity and bilateral pulmonary infiltrates on imaging may constitute independent risk factors for thromboembolic events despite prophylactic-dose heparin in hospitalized acute COVID-19 patients. Our findings could help clinicians identify the patients who may benefit the most from high (i.e., therapeutic/intermediate) doses of heparin. Larger studies are needed to confirm our findings and to evaluate the risk–benefit ratio linked to increasing heparin dosage in these patients.

## Figures and Tables

**Table 1 viruses-14-00737-t001:** Demographics and clinical characteristics of patients on admission.

	Total (*n* = 208)	VTE (*n* = 37)	Non-VTE (*n* = 171)	*p*-Value
Age, years	77 (66–86)	71 (61–81)	78 (69–86)	**0.004**
Sex				
Female	110 (53%)	15 (41%)	95 (56%)	0.09
Male	98 (47%)	22 (59%)	76 (44%)	-
Body Mass Index	26 (23–31)	29 (27–31)	26 (23–29)	**0.002**
Comorbidity				
Hypertension	131 (63%)	28 (76%)	103 (60%)	0.08
Diabetes	39 (19%)	7 (19%)	32 (19%)	0.98
Coronary heart disease	11 (5%)	2 (5%)	9 (5%)	0.97
Cerebrovascular disease	25 (12%)	4 (11%)	21 (12%)	0.8
Previous thrombotic events	13 (6%)	3 (8%)	10 (6%)	0.61
Active cancer	30 (14%)	8 (22%)	22 (13%)	0.17
Chronic kidney disease	43 (21%)	7 (19%)	36 (21%)	0.77
Blood group				
A	83 (40%)	19 (51%)	64 (37%)	0.12
B	28 (13%)	5 (14%)	23 (13%)	0.99
AB	12 (6%)	3 (8%)	9 (5%)	0.5
O	85 (41%)	10 (27%)	75 (44%)	0.06
Padua prediction score	4 (3–6)	3 (3–5)	4 (3–6)	**0.03**
≥4	118 (57%)	13 (35%)	105 (61%)	**0.04**
SOFA score	2 (2–3)	3 (2–3)	2 (1–3)	0.06
≥2	158 (76%)	32 (86%)	126 (74%)	0.09
SIC score	2 (2–3)	2 (2–3)	2 (2–3)	0.49
≥4	29 (14%)	5 (14%)	24 (14%)	0.93
PaO_2_/FIO_2_, mmHg	272 (222–314)	258 (195–272)	277 (232–320)	**0.03**
<300	138 (66%)	29 (78%)	109 (64%)	0.09

Data are median (IQR) or *n* (%); *p*-values were calculated by χ^2^ test or Mann–Whitney U test, as appropriate. SOFA = Sequential Organ Failure Assessment. SIC = Sepsis-Induced Coagulopathy. PaO_2_ = Partial pressure of Oxygen. FIO_2_ = Fraction of Inspired Oxygen.

**Table 2 viruses-14-00737-t002:** Laboratory and radiographic findings of patients on admission.

	Total (*n* = 208)	VTE (*n* = 37)	Non-VTE (*n* = 171)	*p*-Value
**Laboratory findings**				
White blood cell count, ×10^9^ per L	7 (5–11)	9 (6–11)	6.9 (5–11)	0.82
>10	60 (29%)	12 (32%)	48 (28%)	0.59
Lymphocyte count, ×10^9^ per L	1.0 (0.7–1.3)	1.1 (0.6–1.6)	1.0 (0.7–1.2)	0.97
<0.8	74 (36%)	14 (38%)	60 (35%)	0.75
Hemoglobin, g/L	124 (110–137)	128 (117–138)	123 (107–137)	0.12
<120 g/L	91 (44%)	11 (30%)	80 (47%)	0.06
Platelet count, ×10^9^ per L	215 (173–284)	208 (177–279)	215 (171–285)	0.82
<100	17 (8%)	1 (3%)	16 (9%)	0.18
Prothrombin time, s	11 (9–16)	13 (10–16)	12 (9–15)	0.15
Activated partial thromboplastin time, s	24 (22–26)	24 (21–25)	24 (22–27)	0.09
D-dimer, ng/mL	298 (169–657)	317 (197–1093)	289 (161–603)	0.32
>500	71 (34%)	14 (38%)	57 (33%)	0.6
Fibrinogen, g/L	4.9 (4.2–5.4)	5.2 (4.7–6.0)	4.9 (4.2–5.4)	0.76
>4.5	135 (65%)	28 (76%)	107 (63%)	0.13
Antithrombin, %	93 (84–103)	95 (85–107)	92 (83–103)	0.25
<70	7 (3%)	2 (5%)	5 (3%)	0.45
Factor VIII, %	196 (146–253)	220 (167–248)	195 (144–254)	0.33
>200	100 (48%)	21 (57%)	79 (46%)	0.24
Von Willebrand factor, %	299 (242–392)	277 (247–473)	299 (242–381)	0.58
>200	197 (95%)	36 (97%)	161 (94%)	0.44
WB hypercoagulable profile	150 (72%)	28 (75%)	112 (69%)	0.23
WB platelet hyper-reactivity	141 (68%)	26 (70%)	113 (66%)	0.62
Creatinine, mg/dL	0.9 (0.7–1.2)	1.0 (0.7–1.2)	0.9 (0.7–1.2)	0.62
>1.2	47 (23%)	9 (24%)	38 (22%)	0.78
Serum ferritin, μg/L	584 (352–931)	663 (476–1050)	575 (335–894)	0.22
>300	169 (81%)	33 (89%)	136 (80%)	0.17
IL-6, pg/mL	56 (27–82)	47 (21–74)	56 (27–82)	0.95
>82	53 (25%)	9 (24%)	44 (26%)	0.86
C-reactive protein, mg/L	66 (28–110)	58 (15–100)	66 (29–110)	0.35
>110	49 (24%)	7 (19%)	42 (25%)	0.46
Procalcitonin, ng/mL	0.11 (0.04–0.36)	0.10 (0.04–0.23)	0.12 (0.05–0.4)	0.96
>0.36	49 (24%)	5 (14%)	44 (26%)	0.11
**Radiographic findings**				
Consolidation	21 (10%)	2 (5%)	19 (11%)	0.29
Ground-glass opacity	49 (24%)	6 (16%)	43 (25%)	0.25
Bilateral pulmonary	94 (45%)	25 (68%)	69 (40%)	**0.02**
Infiltration				

Data are median (IQR) or *n* (%); *p*-values were calculated by χ^2^ test or Mann–Whitney U test, as appropriate. WB = Whole Blood. IL-6 = Interleukin-6.

**Table 3 viruses-14-00737-t003:** Risk factors associated with VTE.

	Univariable RR (95% CI)	*p*-Value	Multivariable RR (95% CI)	*p*-Value
**Demographics and clinical characteristics**				
Age, years				
38–71	1 (ref)		1 (ref)	
72–82	0.89 (0.47–1.69)	0.72	-	-
83–100	0.47 (0.22–1.02)	0.05	0.52 (0.20–1.05)	0.07
Female sex (vs. male)	0.61 (0.33–1.10)	0.09	0.68 (0.29–1.12)	0.11
Body Mass Index (BMI)				
<30	1 (ref)		1 (ref)	
≥30	**1.85 (1.04–3.31)**	**0.04**	**1.75 (1.02–3.36)**	**0.04**
Comorbidity present (vs. not present)				
Hypertension	1.83 (0.91–3.67)	0.08	1.78 (1.01–3.73)	0.09
Diabetes	1.01 (0.48–2.13)	0.98	-	-
Coronary heart disease	1.02 (0.28–3.71)	0.97	-	-
Cerebrovascular disease	0.89 (0.34–2.29)	0.8	-	-
Previous thromboembolic event	1.32 (0.47–3.74)	0.61	-	-
Active cancer	1.64 (0.83–3.23)	0.17	-	-
Chronic kidney disease	0.90 (0.42–1.90)	0.77	-	-
Blood group O (vs. non-O)	0.54 (0.27–1.05)	0.06	0.48 (0.22–1.15)	0.08
Padua prediction score				
<4	1 (ref)		1 (ref)	
≥4	**0.41 (0.22–0.77)**	**0.04**	**0.54 (0.17–0.81)**	**0.04**
SOFA score				
<2	1 (ref)		1 (ref)	
≥2	2.03 (0.83–4.92)	0.09	2.07 (0.66–5.01)	0.13
SIC score				
<4	1 (ref)			
≥4	0.96 (0.41–2.27)	0.93	-	-
PaO_2_/FIO_2_, mmHg				
≥300	1 (ref)		1 (ref)	
<300	1.84 (0.89–3.81)	0.09	1.91 (0.77–3.96)	0.11
**Laboratory findings**				
White blood cell count, ×10^9^ per L				
≤10	1 (ref)			
>10	1.18 (0.64–2.2)	0.59	-	-
Lymphocyte count, ×10^9^ per L				
≥0.8	1 (ref)			
<0.8	1.1 (0.6–2.01)	0.75	-	-
Hemoglobin, g/L				
≥12	1 (ref)		1 (ref)	
<12	0.54 (0.28–1.04)	0.06	0.47 (0.31–1.18)	0.07
Platelet count, ×10^9^ per L				
≥100	1 (ref)			
<100	0.31 (0.05–2.14)	0.18	-	-
Prothrombin time, s				
≥12	1 (ref)			
<12	0.37 (0.21–1.23)	0.31	-	-
Activated partial thromboplastin time, s				
<26	1 (ref)			
≥26	0.55 (0.26–1.18)	0.17	-	-
D-dimer, ng/mL				
≤500	1 (ref)			
>500	1.17 (0.64–2.14)	0.6	-	-
Factor VIII, %				
≤200	1 (ref)			
>200	1.42 (0.79–2.56)	0.24	-	-
Von Willebrand factor, %				
≤200	1 (ref)			
>200	2.01 (0.3–13.33)	0.44	-	-
WB hypercoagulable profile (vs. other profiles)	1.51 (0.76–3.02)	0.23	-	-
WB platelet hyper-reactivity (vs. other profiles)	1.17 (0.62–2.23)	0.62	-	-
Creatinine, mg/dL				
≤1.2	1 (ref)			
>1.2	1.10 (0.56–2.17)	0.78	-	-
Serum ferritin, μg/L				
≤300	1 (ref)			
>300	1.90 (0.72–5.06)	0.17	-	-
IL-6, pg/mL				
≤82	1 (ref)			
>82	0.94 (0.47–1.86)	0.86	-	-
C-reactive protein, mg/L				
≤110	1 (ref)			
>110	0.76 (0.35–1.62)	0.46	-	-
Procalcitonin, ng/mL				
≤0.36	1 (ref)		1 (ref)	
>0.36	0.51 (0.21–1.23)	0.11	0.71 (0.13–1.49)	0.22
**Radiographic findings**				
Bilateral pulmonary infiltration (vs. other)	**2.33 (1.12–4.85)**	**0.02**	**2.41 (1.09–5.69)**	**0.04**

OR = risk ratio. SOFA = Sequential Organ Failure Assessment. SIC = Sepsis-Induced Coagulopathy. PaO_2_ = Partial pressure of Oxygen. FIO_2_ = Fraction of Inspired Oxygen. WB = Whole Blood. IL-6 = Interleukin-6.

## Data Availability

The data presented in this study are available on request from the corresponding author.

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
