# Peer review of "Risk Factors of Venous Thromboembolism in Noncritically Ill Patients Hospitalized for Acute COVID-19 Pneumonia Receiving Prophylactic-Dose Anticoagulation"

_viruses, 2022, doi:10.3390/v14040737_

Round 1
Reviewer 1 Report
Dear Authors
interesting work in a common question for clinicians in daily practice regarding the anti-coagulation and COVID-19.
well organized prospective study with well presented results.
if more studies in different ethnicities with non critically ill COVID-19 patients exist, should be presented
Author Response
We thank the Reviewer for this interesting comment. A lower thromboembolic risk has been reported in Asian than in Western populations (1). Nevertheless, the data published in the literature so far do not suggest to modify the heparin prophylaxis according to the ethnic group.
- Iba T, Connors JM, Spyropoulos AC, Wada H, Levy JH. Ethnic differences in thromboprophylaxis for COVID-19 patients: should they be considered? Int J Hematol. 2021 Mar;113(3):330-336.
Reviewer 2 Report
In their well-written and referenced paper, the authors address a crucial highly practical topic. For an acute viral potentially fatal disease, i.e SARS-Cov-2 hypoxemic pneumonia, which prognosis is notably dependent on acute thromboembolic venous events (TEVE), they try to find out some factors predictive of high coagulation activation leading to a higher TEVE occurrence. Of course, a conventional one fit-all LWMH prophylactic low dose has been previously shown significantly but partially effective. Highlighting predictive factors might lead to promote studies evaluating an early adaptative anticoagulant (LWMH) dose (low, intermediate, high) use to improve the proven SARS-Cov-2 hypoxemic pneumonia prognosis without excess bleeding.
In this italian, prospective, monocentric study, including 208 consecutive patients with hypoxemic SARS-Cov-2 pneumonia admitted in a non Intensive Care ward, the authors assess a wide range of clinical and biological factors (notably markers of coagulation activation). In a short median survey (10 days), TEVE (DVT, PE) occurred in 18% despite systematic prophylactic LWMH. TEVE have been diagnosed by ultrasonography (US) or CT pulmonary angiography (CTPA).
Out of these factors, high BMI and large pulmonary infiltrates seemed to be the only two simple significant predictive factors (or markers ?) of TEVE occurrence. Surprinsingly, factors usually responsible for TEVE, mainly older age, history of TEVE or active cancer were not significantly related to. Nevertheless, median age was particularly high in this study (median age 77 yo). Among risk scores (PPS, SOFA, SIC), PPS (clinical risk score of TEVE risk in hospitalized patients) was the only one to be significantly effective to predict TEVE occurrence. No biological parameters (blood count, coagulation and inflammation profile, especially very high levels of D-dimer i.e > 3000 ng/ml) provided any significant information to delineate a higher risk of TEVE.
To note, male/female ratio was close to 1 whereas it is usually close to 3 in studies addressing the topic of severe SARS-Co-2 pneumonia. Beside CRP and D-Dimer levels relatively unusual (units used in table 2 are a little bit disturbing. Respectively mg/l and ng/ml seem much more understandable) at the admission, other patients characteristics were similar to those reported in previous studies addressing the same topic.
Finally, the authors should be thanked for this valuable study. They should give some explanations about few points before any consideration of their paper to be published :
- TEVE diagnosis : It is unclear in which circumstances US and CTPA have been carried out.
- Pulmonary infiltrates : It is unclear if diagnosis of extent has been made indifferently by chest X-Ray and/or by CT.
- CRP and D-dimer units in table 2 : Probably, there is a mistake in the unit of D-dimer unit reported in this table (ng/l instead of ng/ml). D-dimer > 500 means > 5000 ng/ml ? If not, D-dimer > 500 ng/ml is an inappropriate cut-off in an old-aged population.
Instead of a single measurement at the admission, would the monitoring of d-dimer levels, coagulation and inflammation profile during the hospitalization stay have been useful as a means of anticipating TEVE ?
Author Response
TEVE diagnosis: It is unclear in which circumstances US and CTPA have been carried out.
Answer: Diagnostic tests were applied if thrombotic complications were clinically suspected. Thank you.
Pulmonary infiltrates: It is unclear if diagnosis of extent has been made indifferently by chest X-Ray and/or by CT.
Answer: Pulmonary infiltrates were diagnosed indifferently by chest X-Ray or by CT. Thank you.
CRP and D-dimer units in table 2: Probably, there is a mistake in the unit of D-dimer unit reported in this table (ng/l instead of ng/ml). D-dimer > 500 means > 5000 ng/ml? If not, D-dimer > 500 ng/ml is an inappropriate cut-off in an old-aged population.
Answer: D-dimer unit reported both in the text and in Table 2 is wrong. It was reported “ng/L” instead of “mg/L”. The manuscript has been amended accordingly. Thank you.
Instead of a single measurement at the admission, would the monitoring of d-dimer levels, coagulation and inflammation profile during the hospitalization stay have been useful as a means of anticipating TEVE?
Answer: We agree with the Reviewer’s comment. Monitoring coagulation as well as inflammation profile during the hospitalization could have provided more comprehensive data on predictive risk factors for VTE. Unfortunately, for most of the parameters considered in our study, it was not possible to carry out repeated evaluations during hospitalization. We have mentioned this aspect among the limitations of the study. Thank you.